# The Role of Organic Small Molecules in Pain Management

**DOI:** 10.3390/molecules26134029

**Published:** 2021-07-01

**Authors:** Sebastián A. Cuesta, Lorena Meneses

**Affiliations:** Laboratorio de Química Computacional, Facultad de Ciencias Exactas y Naturales, Escuela de Ciencias Químicas, Pontificia Universidad Católica del Ecuador, Av. 12 de Octubre 1076 Apartado, Quito 17-01-2184, Ecuador; sebastian_cuesta@yahoo.com

**Keywords:** anti-inflammatory drugs, QSAR, pain management, cyclooxygenase, multitarget drug, cannabinoid, neuropathic pain

## Abstract

In this review, a timeline starting at the willow bark and ending in the latest discoveries of analgesic and anti-inflammatory drugs will be discussed. Furthermore, the chemical features of the different small organic molecules that have been used in pain management will be studied. Then, the mechanism of different types of pain will be assessed, including neuropathic pain, inflammatory pain, and the relationship found between oxidative stress and pain. This will include obtaining insights into the cyclooxygenase action mechanism of nonsteroidal anti-inflammatory drugs (NSAID) such as ibuprofen and etoricoxib and the structural difference between the two cyclooxygenase isoforms leading to a selective inhibition, the action mechanism of pregabalin and its use in chronic neuropathic pain, new theories and studies on the analgesic action mechanism of paracetamol and how changes in its structure can lead to better characteristics of this drug, and cannabinoid action mechanism in managing pain through a cannabinoid receptor mechanism. Finally, an overview of the different approaches science is taking to develop more efficient molecules for pain treatment will be presented.

## 1. Introduction

Inflammation is a very complex self-defense biological process to protect the body against harmful stimuli including pathogens, physical injury, or contact with irritant substances [1]. There are five classical signs of inflammation, i.e., pain, redness, swelling, heat, and loss of function. Although the function of an inflammatory process is to eliminate the cause of injury and heal damaged tissue by clearing dead cells, sometimes this response is too aggressive, causing excessive pain and incapacity [2]. In these cases, an anti-inflammatory drug is needed to ameliorate symptoms and allow the person to continue a normal life. Inflammation is involved in many disorders and complex diseases, including metabolic syndrome, autoimmune, depression, and neurodegenerative diseases [1,3,4]. In first-world societies, the excessive nutrient storage caused by food security and lack of physical activity have stressed humans’ metabolic pathways, causing diseases. The metabolic syndrome is composed of a group of conditions including dyslipidemia, hypertension, obesity, and elevated glucose levels, causing diabetes and atherosclerotic cardiovascular disease [5]. It is known that inflammation plays a pivotal role in the pathogenesis of the metabolic syndrome. Although the mechanism is not yet fully understood, it is believed that reactive oxygen species, free fatty acid intermediates, and adipose tissue dysregulation promote inflammation through high levels of proinflammatory adipokines and low levels of anti-inflammatory adiponectines [4,6,7,8]. The inflammatory process is so complex that whole animal models are needed. In this sense, the zebrafish has emerged as a key tool to study inflammatory diseases. Zebrafish present receptors, mediators, and inflammatory cells similar to mammals and humans making them suitable animal models to study new anti-inflammatory agents and their mechanisms [9]. There are different types of anti-inflammatory agents, including small molecules, peptides, and antibodies. In this review, there will be a focus on small molecules for anti-inflammatory treatments as they have been the center of traditional medicine. Small molecule drugs are compounds with low molecular weight that can easily enter the body and modulate biochemical processes to treat medical conditions [10]. Cyclic small molecules including naturally occurring and synthetic heterocyclic and polycyclic compounds are key to produce new drugs to target the inflammation process [11].

Nature represents one of the greatest sources of anti-inflammatory agents. Recent investigations have found novel anti-inflammatory agents from natural sources, including cyanobacterial extract [12], a norditerpenoid from the Hainan soft coral *Sinularia siaesensis* [13], and peptides extracted from the adzuki bean [14]. Traditional Chinese medicine has used herbs such as *Andrographis paniculata* to treat fever, coughs, and other cold symptoms. The active ingredient andrographolide extracted from this herb has been tested as a multitarget inflammatory drug, showing good results against multiple sclerosis, some respiratory diseases, and osteoarthritis [15]. Complex ancient preparations used for centuries, such as the Shiyifang Vinum containing 13 herbs, have been employed to treat pain and inflammation [16]. One of the first anti-inflammatory treatments described by ancient civilizations, such as Egyptians, Greeks, Sumerians and Chinese, was the use of the willow bark [17,18,19]. Preparations of Salix species have been used for centuries to alleviate pain and to treat fever and rheumatic conditions [17,18,19,20]. Its principal active ingredient is called salicin, which is used as a prodrug of salicylate [21]. Willow bark is considered one of the first examples of modern drug development from herbal plants. Salicylic acid was obtained from hydrolysis followed by oxidation of salicin. An acetylated derivative of salicylic acid became one of the most important drugs in the world, aspirin (Figure 1) [17]. The effectiveness and safety profile of herbal medicines such as willow bark extract is of great interest. In this sense, the ethanolic extract showed to be effective and safe to treat low back pain [21,22], and it showed a moderate analgesic effect against osteoarthritis [17]. The action of willow bark extract as an anti-inflammatory agent was compared to celecoxib and acetylsalicylic acid, and it was found to be as effective as these drugs [23]. The extract contains only 24% salicin, which suggests other components of the extract such as flavonoids are helping to increase its effectiveness [20].

### Prostaglandins and the Cyclooxygenase Anti-Inflammatory Pathway

The main target of anti-inflammatory drugs (NSAIDs) is the cyclooxygenase (COX) pathway composed of two isoforms of the enzyme, i.e., cyclooxygenase-1 (COX-2) and cyclooxygenase-2 (COX-2). Both isoforms have an alpha carbon RMSD of only 0.9 Å. Cyclooxygenases are key in the lipid signaling pathway, being the first and rate-dependent step in prostaglandin and thromboxane synthesis [24,25]. Therefore, NSAIDs work by inhibiting the COX pathway and preventing the synthesis of prostaglandins [1]. In the body, arachidonic acid (AA) is transformed by COX-1 and COX-2 in lipid signaling to prostaglandin and thromboxane to mediate inflammatory processes [26]. COX-1 is expressed in all the body, including the kidney and stomach [27], while COX-2 is only expressed at the site of inflammation [28,29].

Functions of prostaglandin other than mediating the inflammation processes are to protect the gastric mucosa and stimulate platelet aggregation. Therefore, reversible COX-1 inhibition may be used as an antiplatelet agent helping patients with cardiovascular conditions when aspiring is not sufficient. Selective COX-1 inhibitors may be safer than nonselective inhibitors associated with higher risks of upper gastrointestinal bleeding [30] and selective COX-2 inhibitors that are linked to cardiovascular effects [31,32,33].

The active site of COX-1 enzyme is formed by Val116, Arg120, Tyr348, Val349, Leu352, Tyr355, Leu359, Phe381, Leu384, Tyr385, Trp387, Phe518, Ile523, Ala527, Ser530 and Leu531 [24]. Its difference with the COX-2 active site is the presence in COX-2 of Ile523 instead of Val509 and Arg499 instead of His513, which makes the COX-2 active site 20% bigger and with a hydrophilic side chamber [34,35,36]. COX-2-selective inhibitors take advantage of the side pocket and bind in a different mode compared to nonselective compounds, making an additional salt bridge and three extra hydrogen bonds [24]. NSAIDs inhibit COX enzymes in a reversible competitive manner or in a slow tight-binding way depending on the speed and efficiency of each molecule in displacing water molecules inside the pocket and forming hydrogen bonding [37].

Structural and functional information about COXs has been widely studied, identifying structural features key for binding [25] and using different techniques such as saturation transfer difference NMR spectroscopy (STD) [38]. To fit the active site, the basis of an NSAID chemical structure is generally composed of an aromatic ring and an acidic group. While the aromatic ring makes hydrophobic interactions with the pocket, the acidic group forms hydrogen bonds with Arg120 and Tyr355 [24]. In the study of Viegas et al., known COX-1 and COX-2 inhibitors were evaluated, finding a good correlation between experimental crystallographic structure and STD signal [38]. Ibuprofen, diclofenac, and ketorolac bind in a similar way to COX-2, where the ligand moieties form tighter interactions towards Arg-120 and Tyr-355 than towards Ser-520 and Tyr-385 [38].

The importance of the COX enzymes in cancer-related inflammation has been widely studied [26]. Some types of cancer such as epithelial ovarian cancer are reported to overexpress COX-1. In this scenario, selective COX-1 inhibitors may be useful as clinically proven to detect cancer in an early state (imaging agents) but also as therapeutic agents [39]. The similarity between COX-1 and COX-2 makes it a challenge to synthesize selective inhibitors [40]. The link between inflammation and cancer occurs in the inflammatory pathway and includes proinflammatory agents such as cytokines. Studies have also shown upregulation of COX-2 during cancer, and COX-2 has been associated with some neurodegenerative diseases [40,41].

## 2. Nonsteroidal Anti-Inflammatory Drugs (NSAIDs)

The most prescribed family of anti-inflammatory drugs are NSAIDs, accounting for 5–10% of total prescriptions [42,43,44,45]. Some of them are over-the-counter drugs, which increases their usage [46]. Ibuprofen market alone was valued at USD 294.4 million in 2020 and is expected to reach USD 447.6 million by the end of 2026 [47,48]. NSAIDs are the most cost-effective initial therapy for inflammation and pain relief, including sports injuries, arthritis and headaches [49,50,51]. Although NSAIDs are somewhat effective for spinal pain and other acute painful conditions, there is an urgent need for new therapies to treat these medical conditions [52,53].

NSAIDs’ most common side effects include renal, hepatic, gastrointestinal, and cardiovascular reactions [46,53,54,55,56]. These side effects are known to be enhanced when the patient presents medical conditions such as diabetes, obesity and hypertension [53]. The most common and severe one is gastrointestinal (GI) bleeding and ulceration [57,58,59]. This side effect is mostly attributed to inhibition of COX-1, although there are several other factors involved such as the interaction of NSAID with phospholipids [46]. This is the major impediment to the use of this type of medication in chronic patients [60]. To ameliorate the GI effect, gastroprotective drugs like proton pump inhibitors are used together with NSAIDs [53]. In this sense, more than improving the potency of NSAIDs, what is important regarding this type of drug is to enhance gastrointestinal safety [34,45].

Ibuprofen is considered one of the first and best options when talking about NSAIDs. It was created by a group of scientists from Boots company in 1960 [61], proving to be more effective than its predecessors and, in turn, causing fewer side effects. Ibuprofen belongs to the family of propionic acid derivatives. This family is characterized by moderate efficacy, having analgesic, antipyretic and anti-inflammatory action. Its main difference from other propionic acid derivatives lies in its pharmacokinetic characteristics [62]. It is used in antirheumatic treatments, sports injuries, and menstrual cramps. Although it has been on the market for more than 50 years, there are still ways to change this molecule to enhance its properties and activity. One way is by increasing the number of rotatable bonds to less than or equal to ten, which will increase bioavailability [63]. Another approach reported by Kleemiss et al. is to replace one carbon atom with a silicon one, creating the so-called sila-ibuprofen (**1**; Figure 2) [64,65]. The silicon atom is considered a carbon bioisostere that could enhance biological activity while reducing toxicity. Due to its bigger atomic radius, adding a silicon atom lengthens a single bond by around 20% [65]. Silicon atoms change lipophilicity values, which can improve solubility in some cases and enhance membrane penetration in others, altering potency and selectivity. As silicon is more electropositive than carbon, hydrogen bonding can be enhanced [65].

Some of the drawbacks of ibuprofen’s properties are its low solubility in physiological media, high melting point, and high melting enthalpy, which make the production of intravenous formulations challenging [64]. By changing the tertbutyl carbon for silicon, its melting point is reduced by 30 °C (from 75 to 45 °C) and its melting enthalpy is reduced by around 10 kJ/mol. The solubility of sila-ibuprofen is enhanced, going from 21 mg/L for ibuprofen to 83 mg/L. Obtaining insights into the difference in their chemical structure, it was noted that the C–Si change produces an increase in electron density of 7%, a longer distance to the hydrogen atom (0.374 Å), and a change of 0.68° in the Si–C–H angle. A difference in the electrostatic potential is also noticed, as it changes from positive to negative around the silicon atom [64]. Free energy perturbation and experimental results indicate COX-1 and COX-2 inhibition, and a low toxicity profile of the drug was maintained [64].

Another change that was shown to enhance the pharmacological profile of a drug is to form a conjugate with a saccharide. Sodano and coworkers synthesized and evaluated a paracetamol–galactose conjugate as a prodrug [66]. Their results revealed the conjugate improved the pharmacodynamic and toxicological profile of the drug. In this regard, the conjugate presented higher stability in human serum and reduced in vitro metabolism, as the conjugate after was able to be found after 2 h, which does not happen when using paracetamol alone. As paracetamol is slowly released from the conjugate, a longer analgesic effect was found after oral administration lasting up to 12 h being able to treat neuropathic pain such as hyperalgesia. Moreover, the hepatotoxicity was significantly reduced compared to paracetamol [66].

Changing functional groups is also an interesting approach to increase the safety profile of a drug. Some studies suggest that carboxylic acid is key for binding, but it is also linked to gastrotoxicity. In this sense, oxetan-3-ol and thietan-3-ol appear to be interesting bioisosteres in COX inhibitors. As oxetane is a carbonyl isostere, carboxylic acid may be exchanged for oxetan-3-ol. Results showed this isostere improves brain penetration and can be used for CNS drug design [67]. Furthermore, linking furoxan and furazan groups with ibuprofen produces compounds **2** and **3** (Figure 2) with better gastrotoxicity properties without changing the anti-inflammatory effect [27].

A lot of work has been conducted in designing new COX inhibitors [68]. QSAR modeling is the most used and powerful approach that allows correlating chemical modifications in a molecule with its biological activity [69]. This approach has been applied to find new treatments for Alzheimer’s disease [70], malaria [71,72], diabetes [73,74], cancer [75,76,77], and HIV [78]. Based on the anti-inflammatory activity of pyrazolo[1,5-a]pyrimidine, 2,5-diarylpyrazolo[1,5-a]pyrimidin-7-amines, new compounds were synthesized. Eleven compounds (**4**–**14**; Figure 2) showed interesting anti-inflammatory properties compared to indomethacin (Table 1) [2]. As shown in Table 1, all compounds except for **7** achieve more than 50% inhibition after 4 h. Furthermore, compound **9** presents only 3.5% less inhibition than indomethacin. An enhanced activity is achieved when R is a chlorine atom. For R’, it was observed that H and F diminished activity while Cl, CH3, and Br increased it.

In another study, Harrak et al. designed 4-(aryloyl)phenyl methyl sulfones as anti-inflammatory compounds against COX-1 and COX-2 [49]. Molecular modeling results showed how the methylsulfone group in the studied compounds fit the COX-2 pocket, making hydrogen bonds with Arg120, Ser353 and Tyr355. *N*-Arylindole (**15**; Figure 2) was found to be the most potent and selective COX-2 inhibitor (COX-1/COX-2 ratio of 262), having greater anti-inflammatory activity than ibuprofen in vivo [49]. Computational studies at the B3LYP/6-31G(d) level revealed the optimized dipole moment of the structures related to COX-2 binding (R^2^ = 0.81). This happens as the dipole moment performs a pivotal role in aligning to the receptor’s binding pocket. Looking at the binding mode in COX-2, the methyl sulfone group of the studied molecules aligns to the carboxylic group of the crystallographic structure of flurbiprofen. The pose of the ligands allows forming hydrogen bonds with Arg120, Ser353 and Tyr355 and van de Waals contacts with Val349, Phe518 and Leu352 [49]. Differences were found between the position of the indole ring compared to the pyrrole and pyridine groups of similar compounds, which led to a more efficient π–π stacking of the indole ring interacting with Phe518. Therefore, the conjunction of a methylsulfone and an aryl group is enough to produce interesting anti-inflammatory activity [49].

Yamakawa et al. proposed that gastric lesions are related to membrane permeabilization. Therefore, loxoprofen derivatives with lower membrane permeabilization should produce fewer gastric lesions. Loxoprofen is used in Japan and is considered safer than indomethacin [79,80]. Synthesized compounds **16** and **17** (Figure 2) have lower membrane permeabilization and indeed produced fewer gastric lesions compared to loxoprofen but with equivalent activity [42]. In a subsequent study, Yamakawa studied the properties and designed analogs of 2-fluoroloxoprofen, where **18** (Figure 2) presented an equivalent ulcerogenic effect with an enhanced potency [81]. Other types of compounds that possess low ulcerogenicity compared to indomethacin are 5-substituted-1-(phenylsulfonyl)-2-methylbenzimidazole derivatives, where compounds **19**, **20** and **21** (Figure 2) also presented good anti-inflammatory properties in in vivo assays, making them good anti-inflammatory candidates [60].

Uddin and coworkers created selective COX-1 inhibitors by taking out the SO_2_CH_3_ group from known COX-2 inhibitor rofecoxib. Starting from the 3,4-diarylfuran-2(5H)-one structure, several fluorinated compounds were designed, synthesized, and tested as COX-1 inhibitors using a structure-activity relationship [39]. It was found that different positions of different functional groups such as trifluoromethyl, halogens, phenoxy, alkyl, alkoxy, and thioalkyl influence COX inhibition and potency. As observed in compound **22** (Figure 2), adding a methoxy group on carbon 4 of one phenyl ring and a fluorine substituent on position 3 or 4 of the other phenyl ring gave the most active compounds with a COX-1 IC_50_ of 0.36 and 0.48 uM, respectively, while their estimated COX-2 IC_50_ is >25 uM. The advantage of having a fluorinated compound is the possibility of using 18F during the synthesis, which will allow the development of prototypes of PET imaging agents [39]. Another approach to find COX-1 inhibitors is the use of protein affinity fingerprints. Hsu and coworkers identified the fingerprint of 19 COX-1 inhibitors and developed a model to screen 62 compounds for new possible COX-1 molecules [82]. Interestingly, although a carboxylate group is present in known COX inhibitors such as ibuprofen or naproxen, that functional group was not found in newly identified COX-1 inhibitors. The structure of **23** (Figure 2) is similar to ketoprofen, suggesting the carboxylic functional group is not needed for activity while the diaryl ketone is [82,83]. Moreover, it has been shown experimentally that flurbiprofen ethyl ester inhibits COX-1 in a similar pose to flurbiprofen with no need for the carboxylic group [37,82].

Vitale et al. found that 3-(5-chlorofuran-2-yl)-5-methyl-4-phenylisoxazole (**24**; Figure 2) is a potent COX-1 inhibitor, where the isoxazole ring and the furanyl group are key for selectivity [30]. Furthermore, a SAR series based on this compound was performed, showing that the incorporation of 5-chlorofuran-2-yl, 4-phenyl and 5-methyl groups on the isoxazole core enhances selective inhibition of COX-1. Two extra changes were shown to increase potency with a slow reversible process: exchanging chlorine with bromine or a methyl group in the furyl core and placing a CF_3_ group instead of the 5-methyl group (**25**, **26**, **27**; Figure 2). For these compounds, COX-1 IC_50_ values of 0.32, 0.33 and 0.18 uM were found. Furthermore, compound **25** was shown to be >1000-fold more potent for COX-1 than COX-2, having a slow reversible interaction with the first one and a fast one with the second [30].

Selective COX-2 inhibitors such as coxibs allow treating inflammation without the gastric effect [53,57,84]. Starting from known selective COX-2 inhibitors and after elucidating their mode of binding by X-ray crystallography or NMR, a pharmacophore model can be built in order to identify the key features for binding and design new molecules that can inhibit COX-2 selectively. The basis of selective inhibition lies in the large Ile523 on the entrance of the side pocket which prevents bulk and rigid functional groups such as the sulfamoyl or sulfonyl from interacting with the COX-1 side pocket [34]. Compounds such as MK-2894 (**28**; Figure 2) [43] have been discovered to be potent COX-2 inhibitors when looking for novel and effective treatments. Several moieties have been tested, including carbocycle [85], imidazoles [86], thiophenes [87], oxazoles [88], pyrazoles [89], furanones [90], pyridazinone [91], *N*-benzoyl-5-sulfonylindole [92], and others. Furthermore, synthesizing amides and esters of known inhibitors such as meclofenamic acid [93] and indomethacin [94] resulted in compounds with interesting COX-2 inhibition properties. In this sense, it has been shown that indomethacin esters and amides are slow, tight-binding COX-2-selective inhibitors eliminating gastric side effects. Furthermore, primary and secondary amides are found to be more potent than tertiary ones. The 4-chlorobenzyl group is very important for potency as the change of this group by 4-bromobenzyl or hydrogen produced inactive compounds. The same occurred when replacing the 2-methyl group on the indole ring with hydrogen [94]. Although compounds **29** and **30** (Figure 2) are very potent, compound **31** (Figure 2) is not very potent; this is due to the lack of the carbonyl group which is suggested to be key for interaction. The presence of the methoxy (**29**) instead of the amino group (**30**) sightly improves the activity (0.02 uM) and has no effect on COX-1 IC_50_ (>66 uM).

Using a 4,9-dihydro-3H-pyrido[3,4-b]indole core, several compounds were designed, with **32** (Figure 2) being a promising substrate-selective inhibitor. Structural studies show a movement of Leu531, which can be key for the substrate-specific inhibition [95]. Harmaline (1,2-benzisothiazol-3(2H)-one-1,1-dioxide) derivatives were also studied as COX-2-selective inhibitors. Compound **33** (Figure 2) presents an activity of 0.09 (SI ¼ 135.9); exchanging the benzene ring with a methyl group to form **34** (Figure 2) increased activity and selectivity to 0.06 mM (SI ¼ 154), and the formation of **35** (Figure 2) increased activity and selectivity to 0.05 mM (SI ¼ 236), which is comparable to celecoxib (IC_50_: 0.05 mM, SI ¼ 296.00) [54]. When exchanging a five-member triazole with a pyrazole, enhanced activity was found. Adding an extra pyrazole produces an active and selective inhibitor comparable with celecoxib. On the other hand, exchanging the dipyrazole with dihydrazone produces an important reduction in potency and selectivity. Therefore, pyrazoles present better selectivity and activity than hydrazone. The most active benzenesulfonamides are the ones bearing dipyrazole, followed by those with pyrazole, triazole and oxadiazole, with activities similar to celecoxib and presenting a low ulcer index [50].

One of the main functions of COX-2 is to oxygenate arachidonic acid, 2-arachidonoylglycerol (2-AG), and arachidonoylethanolamide (AEA). Studies have shown that ibuprofen and mefenamic acid are more potent in inhibiting 2-AG and AEA than AA oxygenation [96]. The mechanism proposed involves the interaction of the drugs with one active site of the homodimer which alters the structure of the second one, impeding the oxygenation of 2-AG and AEA but not of AA. Inhibiting the second active site requires higher concentration than inhibiting only the first one, allowing substrate-selective inhibition [97]. It is reported that COX inactive (R)-profens can inhibit endocannabinoid oxygenation but not arachidonic acid oxygenation in a substrate-selective manner. Results show that smaller substituents are more potent but less selective. In this sense, desmethylflurbiprofen (**36**; Figure 2) exhibits an IC_50_ of 0.11 μM. Aryl groups are better than other groups, and fluriprofen is more potent than ibuprofen [97].

In a novel approach, Bhardwaj et al. used the COX-2 enzyme to produce selective inhibitors in a process called in situ click chemistry. Lead compounds are produced through a [2,3]-cycloaddition inside the active site of COX-2. As shown in Figure 3, 1,4-regioisomers were produced. The SO_2_Me functional group, the orientation of the azide group inside the pocket, the size, and several interactions are key for the synthesis and later inhibition of this selective compound. When adding a series of precursors, COX-2 was able to choose the one that produced the best fit, producing better inhibitors than celecoxib [40].

## 3. Multitarget Drugs

Sometimes, monotherapy is not considered an effective treatment in complex disorders such as cancer, diabetes, infection, or inflammatory conditions [98]. Fixed-dose drug combination is an alternative, although this type of therapy may increase the risk of adverse drug–drug interactions and produce changes in the pharmacokinetic profile of one of the components [99]. Cocrystals have emerged as compounds with improved properties and performance including solubility, incompatibility, and stability. One example is the tramadol-celecoxib cocrystal, which successfully passed phase II clinical trial for the treatment of acute pain [98,100].

The dual inhibition of COX with other enzymes such as 5-lipoxygenase (5-LOX) [33,101], soluble epoxide hydrolase [102], phosphoinositide-3-kinase delta [103], or fatty acid amide hydrolase (FAAH) [31] enhances therapeutic effect. Advantages of this dual inhibition include greater anti-inflammatory effect with reduced side effects [31]. 5-LOX is involved in the metabolism of arachidonic acid. The leukotrienes produced by 5-LOX are important inflammatory mediators and may be related to cancer, cardiovascular diseases, and gastrointestinal reactions. Still, 5-LOX inhibitors do not show anti-inflammatory effects [33].

Hybrid multiligand molecules containing NSAID structures can lead to enhance anti-inflammatory activity by inhibiting different targets inside the inflammation pathway [104,105]. COX-2/5-LOX inhibitors are compounds that offer more efficacy, fewer side effects, and broader anti-inflammatory properties. Hybridization techniques using COX-2 inhibitors and 5-LOX inhibitors present a good strategy to find dual leads. In this sense, celecoxib analogs were designed by replacing the tolyl ring with an *N*-difluoromethyl-1,2-dihydropyrid-2-one, which is a 5-LOX pharmacophore [106]. Similarly, by joining β-boswellic acid with different NSAID molecules via a Steglich esterification, compounds with interesting anti-inflammatory and antiarthritic properties can be obtained [3]. All hybrid molecules have synergistic effects in inflammation and arthritis, with **37** and **38** being the best (Figure 4). The compound **37** is more effective than ibuprofen in inhibiting COX-2 in vivo while achieving the same effect with only a third of the concentration. The antiarthritic activity of the hybrid is better than β-boswellic acids or ibuprofen individually [3].

Gedawy et al. synthesized a series of pyrazole sulfonamide as anti-inflammatory agents where benzothiophen-2-yl pyrazole carboxylic acid derivative **39** (Figure 4) was the best lead found, with an IC_50_ of 5.40 nM for COX-1, 0.01 nM for COX-2, and 1.78 nM for 5-LOX. Its selectivity index towards COX-2 was 344.56 [34].

The phenoxy acetamide, indole, chalcone thiosemicarbazone, and quinoline are important groups in anti-inflammatory agents [47,107,108]. Huang, Qian, and coworkers designed indole-2-amides as anti-inflammatory drugs [33,109]. Results showed that **40**, **41**, **42** and **43** (Figure 4) present interesting anti-inflammatory properties, with **40** being the most promising compound with COX-2 selectivity and dual COX-2/5-LOX activity binding in the same way as coxibs into the COX-2 active site [33].

Dual inhibitors can also be obtained from natural sources. Primin, a quinone extracted from Primula obconica, presents good anti-inflammatory activity. Therefore, related compounds including hydroquinone, benzoquinone, and resorcinol groups were designed and tested for activity against COX-1, COX-2 and 5-LOX. The compounds **44** (2-methoxy-6-undecyl-1,4-benzoquinone) and **45** (2-methoxy-6-undecyl-1,4-hydroquinone) (Figure 4) were found to be dual COX/5-LOX inhibitors. A key structural modification to achieve a dual inhibition is to have a longer alkyl chain in position 6 from 5 to 11 carbons. Adding an acetyl group in the ortho position of **46** (Figure 4) produces compound **47** (Figure 4), enhancing 5-LOX inhibition. Although acetylation negatively affects COX inhibition, it enhances 5-LOX inhibition [57]. In a recent approach, 15-LOX has also been targeted to reduce inflammation. 1,2,4-Triazine-quinoline and benzimidazole–thiazole hybrids have shown a dual COX-2/15-LOX inhibition being better than celecoxib and quercetin in acting on COX-2 and 15-LOX, respectively. The best compound found for 1,2,4-triazine–quinoline was **48** (Figure 4), presenting IC_50_ values of 0.047 µM (COX-2) and 1.81 µM (15-LOX) [110]; the best benzimidazole–thiazole hybrid was **49** (Figure 4), with an IC_50_ of 0.045 µM for COX-2 and 1.67 µM for 15-LOX [111].

In the design of multitarget compounds, **50** ((±)-2-[3-fluoro-4-[3-(hexylcarbamoyloxy)phenyl]phenyl]propanoic acid, ARN2508; Figure 4) was found to be a potent FAAH and COX inhibitor without the gastrotoxic effect. Other potent dual-target compounds found were **58**–**62** (Figure 4). The **50** enantiomer ((S)-(+)) can be considered a FAAH–COX inhibitor with in vivo potency [31]. The compound **51** (Figure 4) presents the same FAAH potency as c-pentyl (IC_50_ 4.8uM), while **52** (Figure 4), c-butyl, presents a loss in potency (48.7 uM), and **53** (Figure 4), c-propyl, showed no potency at all. Regarding COX activity, **51** is equivalent to **54** (Figure 4), **52** is more potent, and **53** is similar to **54**. It seems that the *N*-terminal region of the carbamate group may increase the interaction with FAAH. Adding a methylene group next to the c-hexyl ring (**55**; Figure 4) increases FAAH and COX-1 potency but not COX-2 potency. The compound **56** (Figure 4) inhibits the three enzymes. The compound **57** (Figure 4) is the most active compound, showing that small and branched alkyl groups substantially affect inhibitory activity [31].

Other types of dual inhibitors are the ones inhibiting COX and carbonic anhydrase (CAI) [112]. COX–CAI inhibitors are more effective in treating rheumatoid arthritis than common NSAIDs as they do not increase the risk of oxidative stress in patients [4,113]. Rheumatoid arthritis is an autoimmune inflammatory disease affecting joints, cartilage, and bones [114,115]. Its pharmacological treatment includes the use of common NSAIDs; glucocorticoids (prednisolone); and antirheumatic agents, including aminosalicylates (sulfasalazine), antimalarial drugs (hydroxychloroquine), immunosuppressants/cytostatic drugs (methotrexate), and antirheumatic drugs (aurothiomalate) [116]. Hybrid compounds containing NSAID drugs and sulfonamides and carboxylates moieties can be synthesized as dual COX–CAI inhibitors [117]. In this sense, two molecules (**63** and **64**; Figure 4) were designed and presented interesting properties [113,114]. Furthermore, the 7-coumarine ibuprofen (**65**; Figure 4) hybrid showed a high antihyperalgesic effect and activity against human hCAs, although it was limited to the isoforms IV and XII and did not include IX [114].

Finally, by a covalent link between a sulohydroxamic acid moiety and known NSAIDs via a two-carbon ethyl spacer, prodrugs with anti-inflammatory activity and the ability to release nitric oxide and nitroxyl were synthesized [32]. The esters produced with naproxen and ibuprofen showed greater anti-inflammatory activity than their parent compounds, while indomethacin ester was shown to be a selective COX-2 inhibitor with no ulcerogenic effect [32]. The multitarget drugs described in this section are shown in Figure 4.

## 4. Cannabis

Opioids and NSAIDs are the first options to treat acute pain [118]. Still, those drugs have shown to be not as effective in treating chronic pain, which may be disabling in some cases [119]. The endocannabinoid system involved in pain and inflammation processes has emerged as a promising approach to treat chronic pain [119,120,121]. Cannabinoids acting as positive allosteric CB1 receptor modulators, CB1 agonists, CB2 agonists, and mixed CB1/CB2 agonists have been reported to help in inflammatory and neuropathic pain [122,123,124].

The cannabis plant has been cultivated for hundreds of years and used for medicinal, spiritual, and recreational purposes. The main components in the plant are alkaloids which have been demonstrated to present anti-inflammatory activity [125]. In the cannabis plant, there are more than 540 phytochemicals from 18 different classes. Of those, 100 are phytocannabinoids [120]. Preparations with this plant have been reported to produce anti-inflammatory, anticonvulsant, antianxiety, analgesic, and muscle relaxant effects [126]. Some countries have approved the use of cannabis to treat pain, multiple sclerosis, epilepsy, sleep disturbance, and neurodegenerative diseases. Furthermore, antiseizure effects in Lennox–Gastaut syndrome and Dravet syndrome have been observed in randomized controlled trials [120,127].

The term medical cannabis has arisen in the last years and refers to the use of the cannabis plant to treat different medical conditions as prescribed by a doctor [128,129]. The main use of medical cannabis is for pain management [86]. In this sense, cannabinoids act mainly through cannabinoid receptors, although it has been reported they can also modulate ion channels and enzymes [119]. The most accepted mechanism for the analgesic effect of cannabis is through the reduction of neural inflammation and descending inhibitory pain and the modulation of postsynaptic neuron excitability pathways. Although more than 10,000 scientific articles present “conclusive or substantial evidence” and support the use of cannabis for neuropathic pain, there is a need to carry out long clinical trials to establish its safety, dosage, and indications for medical conditions [119].

Studies have shown cannabinoids act on different targets in the peripheral and central nervous system [130,131]. Their targets include CB1/CB2 receptors, GPR55, GPR18, *N*-arachidonoyl glycine (NAGly) receptor, nuclear receptors, ion channels, and other potential targets in the CNS [130,132]. Furthermore, cannabinoids may also act on γ-aminobutyric acid, serotonergic, adrenergic and opioid receptors that are part of the analgesic pathway [121].

Although cannabinoids are considered the main active ingredients in cannabis, other compounds inside the plant, including terpenoids and flavonoids, may also be involved in the anti-inflammatory and analgesic effect of cannabis [126,133]. The main components of cannabis are cannabidiol (CBD) and tetrahydrocannabinol (THC) [134]. THC was first isolated in the 1960s, and it is responsible not only for the psychoactive effect but also for some of the analgesic and anti-inflammatory properties. Cannabis prohibition started in the 20th century due to its psychoactive properties (ElSohly et al., 2017). Psychotropic effects include euphoria and paranoia that occur due to the activation of CB1 receptors [119].

THC is a partial CB1 and CB2 receptor agonist, while CBD, a nonpsychoactive cannabinoid, has little affinity to both receptors. Studies have shown it is able to antagonize those receptors in presence of THC acting synergically [124]. Obtaining insight into the action mechanism, CBD is a noncompetitive negative allosteric modulator of the CB1 receptor [135]. CBD modulates non-cannabinoid G protein-coupled receptors (GPCRs), ion channels, and peroxisome proliferator-activated receptor (PPARs), affecting the perception of pain [131,136].

Inhaled cannabinoids are rapidly absorbed in the bloodstream. Gastrointestinal absorption is irregular, causing low bioavailability and poor pharmacokinetic profile, although it can be increased with food [137]. This is a limitation for using cannabis in oral formulations [119]. Nabiximols, a mixture of CBD and THC approved in Spain, Switzerland, Australia, Canada, Brazil and other countries, is prescribed to treat spasticity in multiple sclerosis [127]. Currently, there are only two synthetic cannabinoids on the market. Dronabinol and nabilone were accepted to treat chemotherapy-associated nausea and vomiting, but they may also be helpful in treating pain [138].

The lack of interest in cannabinoids was caused by the psychoactive side effect that some of these compounds have. Therefore, scientific efforts should focus on preserving and increasing the analgesic effect of these compounds while reducing their psychoactive effect [119]. In this regard, the medicinal chemistry approach towards the use of cannabinoids to treat medical conditions has focused on the modulation of the cannabinoid receptors CB1/CB2 and two endocannabinoid deactivating enzymes, monoacylglycerol lipase (MGL) and enzymes fatty acid amide hydrolase (FAAH), including allosteric inhibition [139]. Aminoalkylindole is the first different chemotype to be introduced in cannabinoid medicinal chemistry. Compounds such as AM2201 (**66**; Figure 5) and AM678 (**67**; Figure 5) present potent antinociceptive properties, but due to their structure, some of them have been classified as controlled substances. BAY 38-7271 (**68**; Figure 5) is a potent CB1/CB2 receptor agonist with clinical efficacy against severe traumatic brain injury [139].

One approach to eliminate undesirable side effects is to design peripherally restricted agonists. In this sense, SAB378 (**69**; Figure 5) and AZD1940 (**70**; Figure 5) are good candidates due to their interesting properties. The discovery of selective CB2 receptors was important because their presence is peripheral, exhibiting antinociceptive and anti-inflammatory action without the CB1 side effects. AM1241 (**71**; Figure 5) and AM1710 (**72**; Figure 5) were tested in rodent models and found to be successful for treating inflammatory and neuropathic pain [139].

In 29 trials performed between 2003 and 2014, more than 75% of the studies showed a significant analgesic effect in chronic noncancer pain, including neuropathic pain, rheumatoid arthritis, and fibromyalgia [140]. The compounds targeting CB1 and/or CB2 receptors are shown in Figure 5.

## 5. Calcium Channel

Neuropathic pain is a chronic condition affecting 7 to 8% of the population [141,142]. Although NSAIDs and cannabinoids have been used, their effectiveness has not been as expected [121,143]. Neuropathic pain is characterized by discomfort and soreness that is stimulus-independent, affecting the quality of life of the patients [144]. Anticonvulsant drugs including pregabalin and gabapentin have emerged as a new therapy to treat neuropathic pain, showing efficacy in more than ten double-blind clinical trials [142,145]. The analgesic effect of these drugs comes through a different pathway [146]. In 1996, the pregabalin and gabapentin target was found to be the α2-δ subunit of the P/Q type voltage-gated calcium channel [147,148]. Its main mechanism involves returning hyperexcited neurons to a normal state through the reduction of calcium [145,149,150]. Docking studies have shown that pregabalin’s most stable conformation fits in a pocket where 12 interactions including 6 hydrogen bonds were found, including electrostatic interactions with Arg217. It has been shown experimentally that Arg217 is a gating charge carrier essential in pregabalin’s action mechanism, opening the possibility of developing new leads to treat neuropathic pain through calcium channel modulation [149,151,152].

## 6. Conclusions

Pain and inflammation are conditions that affect the quality of life of a high percentage of the human population. Humans have been looking for ways to treat pain and inflammation since ancient times, finding the first treatments in herbal plants such as the willow bark. Salix species contains salicin, which is the main active ingredient helping reduce pain and inflammation in conjunction with other compounds such as flavonoids. Furthermore, this compound was used as starting material to produce one of the most famous anti-inflammatory drugs, aspirin. Although the main mechanisms of action of anti-inflammatory drugs are the COX pathways involving COX-1 and COX-2 enzymes, it is now known the inflammation and analgesic pathways are much more complex and include other proteins such as 5-LOX, FAAH, CAI, ion channels, γ-aminobutyric acid, serotonergic, adrenergic, opioid, and CB1/CB2 receptors that can be and are now tested as potential targets for pain management treatments. In this sense, new types of molecules including cannabinoids are being tested as potential pain treatments with excellent results. Efforts have been focused on designing more potent and safer drugs. In this sense, research has leaned toward the synthesis of cocrystals, COX-selective inhibitors (either COX-1- or COX-2-selective), and dual inhibitors. Thanks to the advances of the technology and through the elucidation of the biological targets and QSAR studies, interesting new molecules have been designed, taking advantage of the pocket size and properties in order to enhance selectivity and reduce side effects. Still, more studies need to be done to produce more effective treatments against more complex conditions including rheumatoid arthritis, neuropathic pain, and inflammation processes present in different types of cancer.

## Figures and Tables

**Figure 1 molecules-26-04029-f001:**
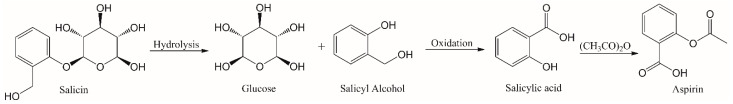
Aspirin synthesis from salicin.

**Figure 2 molecules-26-04029-f002:**
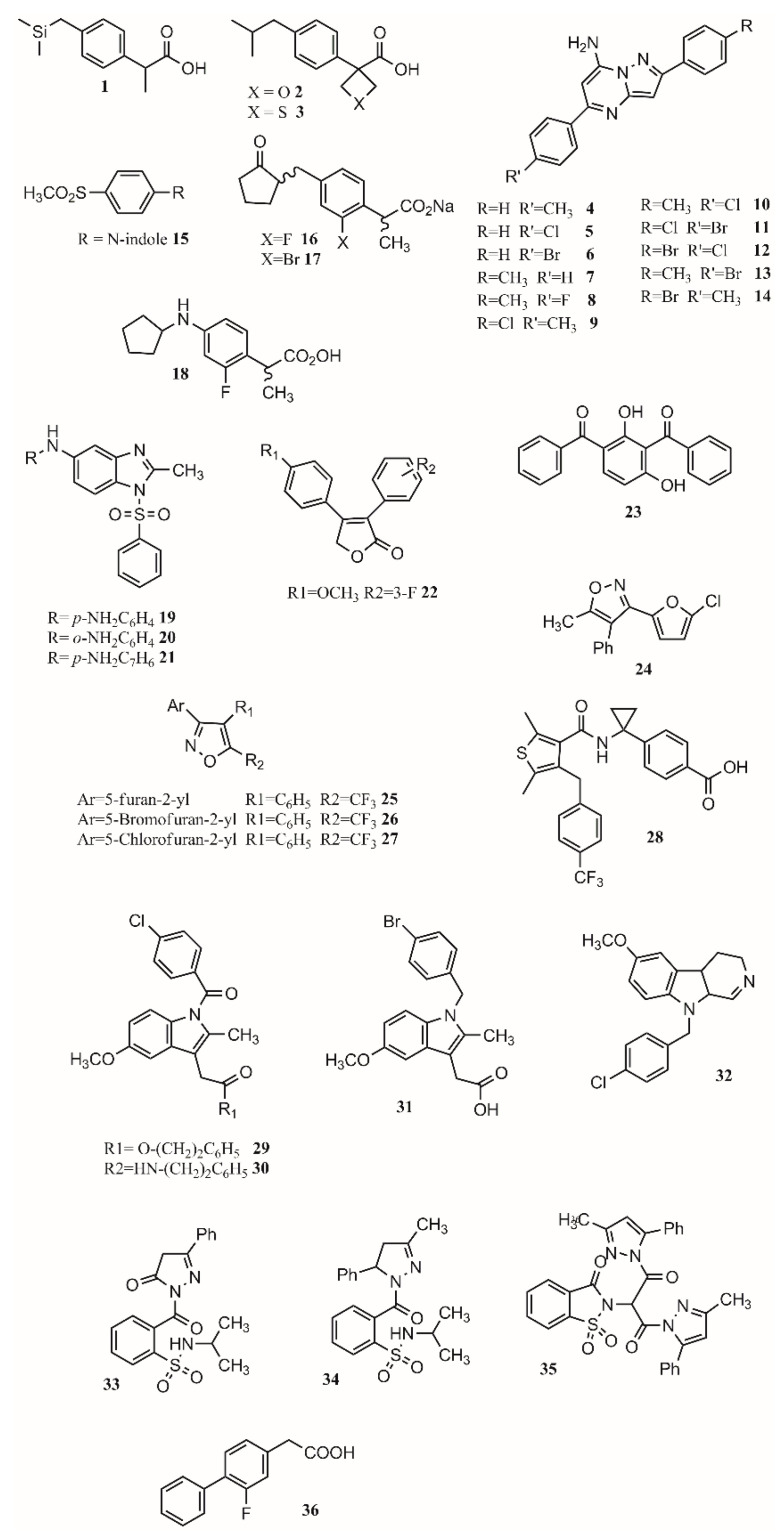
COX inhibitors.

**Figure 3 molecules-26-04029-f003:**
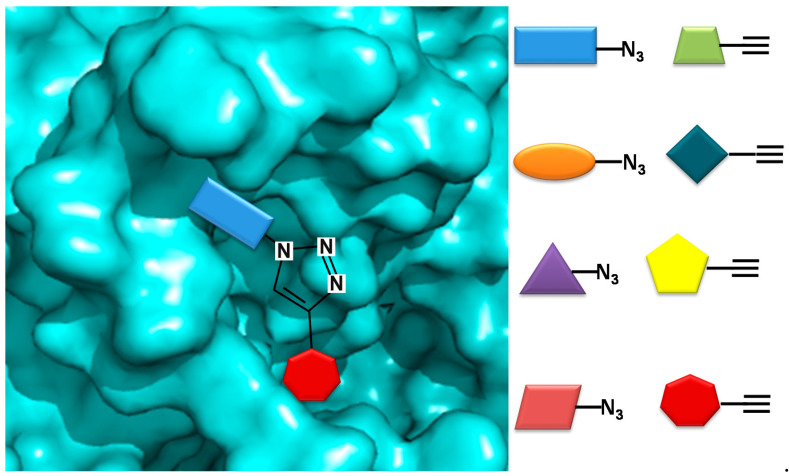
COX-2 inhibitor design using in situ click chemistry.

**Figure 4 molecules-26-04029-f004:**
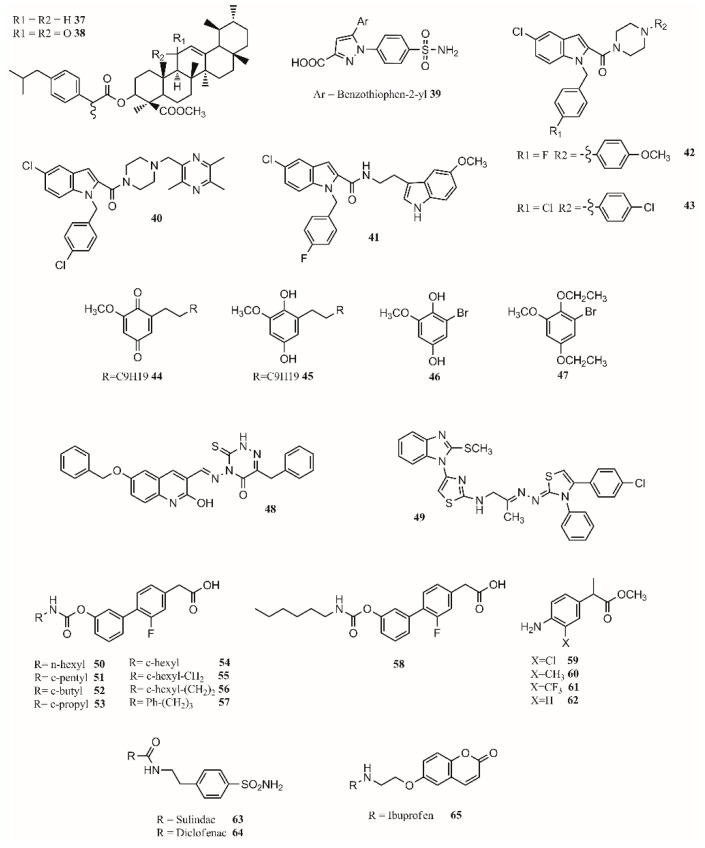
Multitarget drug leads.

**Figure 5 molecules-26-04029-f005:**
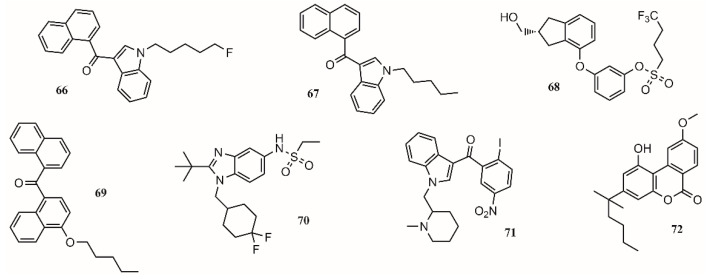
Compounds targeting CB1 and/or CB2 receptors.

**Table 1 molecules-26-04029-t001:** Anti-inflammatory activity after 4 h of compounds **4** to **14** compared to indomethacin.

Compound	Inhibition (%)
**4**	54.76
**5**	51.16
**6**	66.66
**7**	34.52
**8**	64.28
**9**	80.95
**10**	67.85
**11**	76.19
**12**	65.47
**13**	66.66
**14**	64.28
Indomethacin	84.52

## Data Availability

Not applicable.

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
