# Peer review of "The Role of Organic Small Molecules in Pain Management"

_molecules, 2021, doi:10.3390/molecules26134029_

Round 1
Reviewer 1 Report
This is an interesting review about the anti-inflammatory drugs and compounds used for acute pain and inflammation.
Inflammation and cytokines release are process involved in cancer, diabetes, neurodegenerative and chronic disease as metabolic syndrome.
Could you add a paragraph explaining the inflammatory process thataccompanies the metabolic syndrome and the involvement of COX since it
was not mentioned in this article.
In all the paragraphs where the chemical structures of compounds 1 to 70 are
described and mentioned, the corresponding figure should be indicated where
the formulas and chemical structures are found. For example: For COX- inhibitor
compounds (1 to 36) it is worth mentioning Figure 2; for multitarget drugs (37-63),
Figure 4 and for compounds targeting CB1 or CB2 receptors (64-70), Figure 5.
Author Response
This is an interesting review about the anti-inflammatory drugs and compounds used for acute pain and inflammation.
Inflammation and cytokines release are process involved in cancer, diabetes, neurodegenerative and chronic disease as metabolic syndrome.
Could you add a paragraph explaining the inflammatory process that
accompanies the metabolic syndrome and the involvement of COX since it
was not mentioned in this article.
Answer: Thank you for your comments and suggestion. A paragraph has been added to the main text in page as suggested (See paragraph 1).
In all the paragraphs where the chemical structures of compounds 1 to 70 are
described and mentioned, the corresponding figure should be indicated where
the formulas and chemical structures are found. For example: For COX- inhibitor
compounds (1 to 36) it is worth mentioning Figure 2; for multitarget drugs (37-63),
Figure 4 and for compounds targeting CB1 or CB2 receptors (64-70), Figure 5.
Answer: We agree this will make the structures easy to find so we have added the figures throughout the main text.
Reviewer 2 Report
The phrase “Prostaglandins and the Cyclooxygenase Anti-inflammatory pathway” of page 2 must be included in a epigraph.
The title can be misleading. In my opinion, it is very attractive but initially it leads to a broad discussion about the size of the drugs and their relationship with the anti-inflammatory effect. To solve it, you could add some information regarding this question on sheet 8 “Result shows smaller substituents are more potent, but less selective…” and on sheet 9 “most active compound showing small and branched alkyl groups affects substantially inhibitory activity”.
The multitarget drug section is interesting and is a bit lacking considering that it is a review article. Only seven references (86-92) have been used for this section, they are insufficient and many of these references are not current. We must add more works and that have been published in the last three years. Only one of the 126 citations is from 2021. This would have to be considered not only for this section of the article.
Author Response
The phrase “Prostaglandins and the Cyclooxygenase Anti-inflammatory pathway” of page 2 must be included in a epigraph.
Answer: Thanks for the suggestion we have added the phrase as epigraph
The title can be misleading. In my opinion, it is very attractive but initially it leads to a broad discussion about the size of the drugs and their relationship with the anti-inflammatory effect. To solve it, you could add some information regarding this question on sheet 8 “Result shows smaller substituents are more potent, but less selective…” and on sheet 9 “most active compound showing small and branched alkyl groups affects substantially inhibitory activity”.
Answer: Thanks for the comment and sorry for the confusion. When we talk about small molecules, we are talking about the class of drug that includes low molecular weight organics compounds and no other type of drugs such as peptides and antibodies. To avoid future confusion and make the title clear, we have added a couple of sentences at the end of the first paragraph.
The multitarget drug section is interesting and is a bit lacking considering that it is a review article. Only seven references (86-92) have been used for this section, they are insufficient and many of these references are not current. We must add more works and that have been published in the last three years. Only one of the 126 citations is from 2021. This would have to be considered not only for this section of the article.
Answer: Thank you very much for the comment. We start the review at the end of 2020, and that is why we might miss some 2021 research articles. We have added references in the multitarget section (see main text) and added more than 15 new references from 2021 (see reference section).